# ASTRAL: TRAINING PHYSICS-INFORMED NEURAL NETWORKS WITH ERROR MAJORANTS

## ABSTRACT

The primal approach to physics-informed learning is a residual minimization. We argue that residual is, at best, an indirect measure of the error of approximate solution and propose to train with error majorant instead. Since error majorant provides a direct upper bound on error, one can reliably estimate how close PiNN is to the exact solution and stop the optimization process when the desired accuracy is reached. We call loss function associated with error majorant **Astral**: neur**A**l a po**ST**erio**R**i function**A**l **L**oss. To compare Astral and residual loss functions, we illustrate how error majorants can be derived for various PDEs and conduct experiments with diffusion equations (including anisotropic and in the L-shaped domain), convection-diffusion equation, temporal discretization of Maxwell's equation, magnetostatics and nonlinear elastoplasticity problems. The results indicate that Astral loss is competitive to the residual loss, typically leading to faster convergence and lower error (e.g., for Maxwell's equations, we observe an order of magnitude better relative error and training time). The main benefit of using Astral loss comes from its ability to estimate error, which is impossible with other loss functions. Our experiments indicate that the error estimate obtained with Astral loss is usually tight enough, e.g., for a highly anisotropic equation, on average, Astral overestimates error by a factor of 1.5, and for convection-diffusion by a factor of 1.7. We further demonstrate that Astral loss is better correlated with error than residual and is a more reliable predictor (in a statistical sense) of the error value. Moreover, unlike residual, the error indicator obtained from Astral loss has a superb spatial correlation with error. Backed with the empirical and theoretical results, we argue that one can productively use Astral loss to perform reliable error analysis and approximate PDE solutions with accuracy similar to standard residual-based techniques.

## 1 INTRODUCTION

Physics-informed neural networks (PiNNs) can be considered as a solution technique for differential equations (most notably, PDEs) that approximate an unknown solution with a neural network, obtain derivatives with automatic differentiation, and minimize PDE-related loss function with first-order or quasi-Newton method Lagaris et al. (1998), Raissi et al. (2019). One of the principal questions is how reliable PiNNs can be trained and how accurate is the final approximation.

For both of these questions, the appropriate choice of the loss function is crucial. The most widely used option is PDE residual sampled at the set of random points Wang et al. (2023). Adaptively selected points Wu et al. (2023), Zubov et al. (2021) are also used to promote additional error control. For selected problems, variational loss is used E & Yu (2018), Barrett et al. (2022), and weak form with fixed Kharazmi et al. (2019) or adaptive Chen et al. (2023a) test function. For time-dependent problems, special reweighting schemes are also available Wang et al. (2022).

A good loss function is necessary for a highly accurate approximate solution, but not sufficient. One way or another, error analysis needs to be introduced. Many works available that focus on a priori error analysis for PiNNs, e.g., De Ryck et al. (2022), Mishra & Molinaro (2022), De Ryck & Mishra (2022), Gonon et al. (2022), Jiao et al. (2021). In such contributions, authors show that it is possible to reach a given approximation accuracy with a neural network of a particular size. However, a priori error analysis is insufficient to obtain practical error estimation of *trained* neural

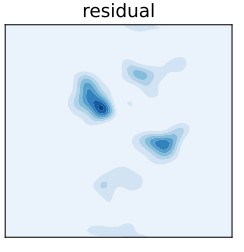 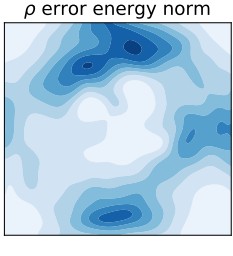 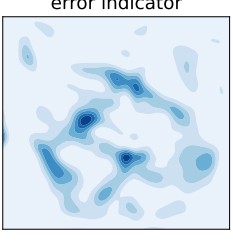 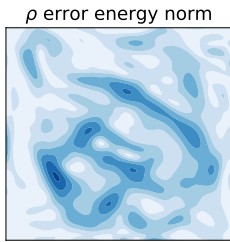

(a) Residual and density of error energy norm         (b) Error indicator and density of error energy norm

Figure 1: Residual, Astral error indicator and density of energy norm for diffusion equation. Residual is completely uncorrelated with error in energy norm, whereas error indicator captures essential details of error distribution. Statistical study shows that for 100 diffusion equations with randomly selected diffusion coefficient residual correlation is $0.22 \pm 0.09$ and Astral error indicator gives correlation $0.82 \pm 0.04$. More examples are available in Appendix C and Appendix D.

network. For that one should resort to a posteriori error analysis for PiNNs, which is also present in literature but to a lesser extent, e.g., Guo & Haghighat (2022), Filici (2010), Hillebrecht & Unger (2022), Berrone et al. (2022), Minakowski & Richter (2023), Roth et al. (2022), Cai et al. (2020). In all these contributions, authors specialize in a particular classical error bound to deep learning PDE/ODE solvers. Namely, Filici (2010) adopts a well-known error estimation for ODEs, based on the construction of related problems with exactly known solution Zadunaisky (1976). Similarly, Hillebrecht & Unger (2022) uses well-known exponential bound on error that involves residual and Lipschitz constant Hairer et al. (1993) and applies a neural network to perform the residual interpolation. Similarly, contributions Guo & Haghighat (2022), Berrone et al. (2022) and Cai et al. (2020) are based on FEM posterior error estimates, and Minakowski & Richter (2023), Roth et al. (2022) are on dual weighted residual estimator Becker & Rannacher (2001).

In the present contribution, we take a different root and propose to address efficient PiNN training and a posteriori error analysis simultaneously. The approach is based on functional a posteriori error estimate Repin (2008), Muzalevsky & Repin (2021) that is approximation-agnostic and, because of that, ideally suited for ansätze based on neural network. The main idea is to derive a tight upper bound on error (error majorant) in a problem-dependent energy norm and use this upper bound as a loss function. This way one can seamlessly combine learning high-quality approximate solutions with a posteriori error control. To summarise, our main contributions are:

1. We introduce Astral – loss function based on error majorant derived with the help of a posteriori functional error estimate.

2. We perform a series of tests of parametric families of diffusion equations (including highly anisotropic cases and equations with large mixed derivatives), convection-diffusion equation, diffusion equation in the L-shaped domain, Maxwell's equation, magnetostatics problem and nonlinear elastoplasticity equation. Our tests indicate that Astral loss is robust, comparable, or more accurate than residual loss, computationally cheaper, and results in a sufficiently tight upper bound.

## 2 MOTIVATION AND THE FIRST LOOK AT THE ASTRAL LOSS

The end goal of physics-informed training is to find an approximate solution for PDE, meaning to obtain small errors. Unfortunately, the error is not computable since the exact solution is not known so a prevailing strategy is to minimize $l_2$ norm of residual sampled at a small set of points Lagaris et al. (1998), Wang et al. (2023). However, it is well known that residuals can fail to provide information about the error. In this section we will provide theoretical and numerical evidence that residual is poorly correlated with error and show that one can define a better loss function.

## 2.1 RESIDUAL AND ERROR FOR A TOY BOUNDARY-VALUE PROBLEM

We consider a simple boundary-value problem

$$\frac{d^2\phi(x)}{dx^2} = 0,\ x \in (-1, 1),\ \phi(-1) = \phi(1) = 0, \tag{1}$$

with trivial exact solution $\phi(x) = 0$. It is easy to construct pathological approximate solutions to this problem that lead to arbitrary relations between error and residual.

Two edge cases are given by the following functions

$$\widetilde{\phi}_1(x, \epsilon) = \epsilon \sin\left(\frac{x}{\epsilon}\right),\ \widetilde{\phi}_2(x, \alpha) = \begin{cases} \alpha(1 + x),\ x \le 0; \\ \alpha(1 - x),\ x > 0. \end{cases} \tag{2}$$

For small $\epsilon$, function $\widetilde{\phi}_1(x, \epsilon)$ has a small amplitude and does not deviate substantially from the exact solution, but simultaneously has a large second derivative, so residual can be made arbitrary large for arbitrary small error.

As noted in Muzalevsky & Repin (2021) function $\widetilde{\phi}_2(x, \alpha)$ is piecewise linear, so the residual is zero for arbitrary $\alpha$ everywhere except $x = 0$, where $\widetilde{\phi}_2(x, \alpha)$ is not differentiable. On the other hand for selecting large $\alpha$ one can make an error arbitrarily large for zero residual. Note, that $x = 0$ is easy to miss since residual is enforced in a finite set of collocation points.

## 2.2 RESIDUAL CORRELATION WITH ENERGY NORM

One can justly argue against the examples in the previous section based on the fact that such pathological cases may not appear in realistic PiNN training.

To evaluate the relation between error and residual in a realistic setting we will consider the diffusion equation

$$-\mathrm{div}\left(a(x, y)\,\mathrm{grad}\,\phi(x, y)\right) = f(x, y),\ x, y \in \Gamma,\ \phi|_{\partial\Gamma} = 0, \tag{3}$$

where $a(x, y)$ is uniformly positive function and $\Gamma = (0, 1)^2$. A natural way to measure error between exact $\phi$ and approximate $\widetilde{\phi}$ solutions is to use energy norm

$$E[\phi - \widetilde{\phi}]^2 = \int dx dy\, a(x, y) \left\| \mathrm{grad}\,\phi(x, y) - \mathrm{grad}\,\widetilde{\phi}(x, y) \right\|_2^2. \tag{4}$$

Integration by parts shows that this error is a $L_2$ scalar product between error $\phi - \widetilde{\phi}$ and the residual, so one may hope that the residual is correlated with it to some extent. The result of training standard physics-informed neural network described in Section 5 is given in the left panel of Figure 1. One can see that the spatial correlation between the error in energy norm and the residual is poor. As shown in Appendix C, where other examples alike are given, the residual is similarly poorly correlated with the ordinary error $\phi - \widetilde{\phi}$. In Appendix E, one can also find that magnitudes are poorly correlated similarly. More specifically, for the anisotropic diffusion equation, residual magnitude is on the order of $10^0$ while error is on the order of $10^{-3}$.

## 2.3 ASTRAL LOSS FOR ELLIPTIC EQUATION

Error in energy norm (4) depends only on a derivative of the exact solution. Suppose in addition to approximate solution $\widetilde{\phi}$ we also introduce an independent variable that approximates the flux of exact solution $\widetilde{F}(x, y) \simeq a(x, y)\mathrm{grad}\,\phi(x, y)$. If we have a reasonably good approximation to the flux, we can rewrite the energy

$$E[\phi - \widetilde{\phi}]^2 \simeq \int dx dy\, a(x, y) \left\| \frac{1}{a(x, y)} \widetilde{F}(x, y) - \mathrm{grad}\,\widetilde{\phi}(x, y) \right\|_2^2. \tag{5}$$

This expression known as an error indicator can be computed explicitly without an unknown exact solution.

The flux, however, is not known and needs to be approximated somehow. One idea, introduced in Lyu et al. (2022) is to split the loss in two terms

$$\int dx dy \left( f(x,y) + \operatorname{div} \widetilde{F}(x,y) \right)^2 + \int dx dy \left\| a(x,y) \operatorname{grad} \widetilde{\phi}(x,y) - \widetilde{F}(x,y) \right\|_2^2. \quad (6)$$

Evidently, if both terms are small in the loss above $\widetilde{\phi}$ approximate exact solution and $\widetilde{F}$ approximate the flux, so error indicator (5) can be computed.

However, the loss above does not provide a good approximation to the magnitude of the energy norm. In Appendix A it is shown (for $D = 1$ case) that this loss can be slightly changed in such a way that it becomes a strict upper bound for error in energy norm that is saturated if and only if $\widetilde{F} \to a \operatorname{grad} \phi$ and $\widetilde{\phi} \to \phi$. Modification is relatively mild and has the following form

$$U = \alpha \int dx dy \left( f(x,y) + \operatorname{div} \widetilde{F}(x,y) \right)^2 + \beta \int dx dy \frac{\left\| a(x,y) \operatorname{grad} \widetilde{\phi}(x,y) - \widetilde{F}(x,y) \right\|_2^2}{a(x,y)}, \quad (7)$$

where $\alpha(a)$ and $\beta(a)$ are some constants that may depend on diffusion coefficient, and some additional parameters (e.g., see (13) and Appendix A).

Functional $U$ above is known as error majorant or a posteriori error estimate of functional type (see Mali et al. (2013)) and is also called Astral loss in this paper. It has many favorable properties, described in the next section.

Using neural networks as approximations for $\widetilde{\phi}$ and $\widetilde{F}$, one can consider $U$ as loss in a physics-informed training. The resulting error indicator is given on the right panel of Figure 1 and it is evident that it provides excellent correlation with error in the energy norm. In Appendix E one can also find that magnitudes have excellent correlation with error. For example, for the anisotropic diffusion equation, error and Astral loss magnitudes are of order $10^{-3}$.

## 3 GENERAL DEFINITION OF ERROR MAJORANTS AND ASTRAL LOSS

Having the discussion from the previous section in mind, we extend the definition of the upper bound and the list of requirements to a more general situation.

Consider PDE in the abstract form

$$\mathcal{A}[\phi, \mathcal{D}] = 0, \quad (8)$$

where $\mathcal{A}$ is a nonlinear operator containing partial derivatives of the solution $\phi$, $\mathcal{D}$ stands for supplementary data such as initial conditions, boundary conditions, and PDE parameters.

For a selected set of PDEs, it is possible to obtain a posteriori functional error estimate of the form

$$E \left[ \widetilde{\phi} - \phi \right] \leq U[\widetilde{\phi}, \mathcal{D}, w], \quad (9)$$

where $\widetilde{\phi}$ is an approximate solution, $w$ are arbitrary free functions from certain problem-dependent functional space, $\mathcal{D}$ is problem data, $U$ is problem-dependent nonlinear functionals called error majorant Mali et al. (2013).

We require majorant to have the following properties: (i) For $\widetilde{\phi} = \phi$ one can find $w$ such that $E \left[ \widetilde{\phi} - \phi \right] = U[\widetilde{\phi}, \mathcal{D}, w]$, that is, the upper bound is saturated; (ii) It is possible to evaluate $U$ efficiently given only problem data and approximate solution; (iii) The upper bound is defined in a continuous sense for arbitrary $\widetilde{\phi}$, $w$ from certain functional spaces, that is, it does not contain information on a particular solution method, grid quantities, convergence, or smoothness properties.

We already demonstrated that functionality with these properties exists for BVP (26). More examples appear in Section 5.

Given error majorant (9) we define Astral loss function and associated optimization problem.

**Astral loss.** *For PDE with functional error majorant $U[\widetilde{\phi}, \mathcal{D}, w]$, and neural network $\mathcal{N}(\mathcal{D}, \theta) = \left( \widetilde{\phi}, w \right)$ with weights $\theta$ that predicts solution $\widetilde{\phi}$ and auxiliary fields $w$ optimize*

$$\min_{\theta} U[\widetilde{\phi}, \mathcal{D}, w] \; s.t. \; \left( \widetilde{\phi}, w \right) = \mathcal{N}(\mathcal{D}, \theta). \quad (10)$$

Because Astral loss is based on the upper bound, we can be sure that the error is smaller than the observed loss. Since bound saturates, we also can be sure that it is possible to reach an exact solution.

# 4    EXAMPLES OF ERROR MAJORANT

Functional error majorants can be derived for various practically relevant equations (see Repin (2008), Neittaanmäki & Repin (2004), Mali et al. (2013) for more details). Here we provide the upper bound for four families of equations that we will later use in Section 5 for experimental comparison of different loss functions.

## 4.1    DIFFUSION EQUATION

Consider diffusion equation

$$-\text{div}\left(\sigma(x,y)\,\text{grad}\,\phi(x,y)\right) = f(x,y),\ x,y \in \Gamma,\ \phi|_{\partial\Gamma} = 0, \tag{11}$$

where $\sigma(x,y)$ is $2 \times 2$ symmetric positive definite matrix, $\Gamma$ is a piecewise smooth domain and $\partial\Gamma$ is the boundary.

Energy norm reads

$$E[\phi - \widetilde{\phi}]^2 = \int dxdy\ \left\| \sigma^{1/2}(x,y)\,\text{grad}\left(\phi(x,y) - \widetilde{\phi}(x,y)\right) \right\|^2, \tag{12}$$

where $\|\cdot\|^2$ is $l_2$ norm applied pointwise, and the upper bound is

$$U_D^2[\widetilde{\phi}, f, \sigma] = \frac{(1+\beta)}{4\pi^2 \inf \lambda_{\min}\left(\sigma(x,y)\right)} \int dxdy\ \left(f(x,y) + \text{div}\,w(x,y)\right)^2$$
$$+ \frac{1+\beta}{\beta} \int dxdy\ \left\| \sigma^{-1/2}(x,y)\left(\sigma(x,y)\,\text{grad}\,\widetilde{\phi}(x,y) - w(x,y)\right) \right\|^2. \tag{13}$$

The derivation is very similar to the one presented in Section 2 and can be found in (Mali et al., 2013, Chapter 3). Note that the upper bound depends on scalar free parameter $\beta$ and vector fields with two components $w(x,y)$.

## 4.2    MAXWELL'S EQUATION

Magnetostatics problem as well as temporal discretization of Maxwell's equation can be presented in the following form

$$\text{curl}\,\mu(x,y)\,\text{curl}\,B(x,y) + \alpha B = f(x,y),\ x,y \in \Gamma,\ \phi|_{\partial\Gamma} = 0, \tag{14}$$

where $\mu(x,y)$ is uniformly positive scalar function, $\alpha$ is non-negative real number, $\Gamma$ is a piecewise smooth domain, $\partial\Gamma$ is the boundary; curl from a vector field reads $\text{curl}\,B(x,y) = \partial_y B_x(x,y) - \partial_x B_y(x,y)$, and curl from a scalar field reads $\text{curl}\,\phi(x,y) = e_x \partial_y \phi(x,y) - e_y \partial_x \phi(x,y)$.

The natural energy norm for Maxwell's equation is

$$E[B - \widetilde{B}]^2 = \int dxdy\ \left( \mu(x,y)\left\| \text{curl}\left(B(x,y) - \widetilde{B}(x,y)\right) \right\|^2 + \alpha \left\| B(x,y) - \widetilde{B}(x,y) \right\|^2 \right). \tag{15}$$

It is convenient to obtain the upper bound for two separate cases Repin (2007) when $\alpha > 0$ and when $\alpha = 0$. For the former case, we have

$$U_{M_1}^2[\widetilde{B}, f, \mu, \alpha, w] = \int dxdy\ \left( \frac{1}{\alpha} \left\| f(x,y) - \alpha\widetilde{B}(x,y) - \text{curl}\,w(x,y) \right\|^2 \right.$$
$$\left. + \frac{1}{\mu(x,y)}\left( y(x,y) - \mu(x,y)\,\text{curl}\,\widetilde{B}(x,y) \right)^2 \right), \tag{16}$$

and for the later case

$$U_{M_2}[\widetilde{B}, f, \mu, \alpha, w] = \frac{1}{2\pi \inf \sqrt{\mu(x,y)}} \sqrt{\int dx dy \, \|f(x,y) - \operatorname{curl} w(x,y)\|^2}$$

$$+ \sqrt{\int dx dy \frac{1}{\mu(x,y)} \left(w(x,y) - \mu(x,y) \operatorname{curl} \widetilde{B}(x,y)\right)^2}. \quad (17)$$

Both error majorants (16) and (17) depend on a scalar auxiliary field $w(x,y)$. Note that $U_{M_2}$ is a straightforward modification of the result from Repin (2007) obtained with the use of Friedrichs's inequality for curl.

## 4.3 CONVECTION-DIFFUSION EQUATION

We consider the initial value problem

$$\frac{\partial u(x,t)}{\partial t} - \frac{\partial^2 u(x,t)}{\partial x^2} + a\frac{\partial u(x,t)}{\partial x} = f(x), u(x,0) = \phi(x), \, u(0,t) = u(1,t) = 0, \quad (18)$$

where $(x,t) \in (0,1) \times (0,T)$ and $a, T$ are positive real numbers. The natural energy norm is

$$E^2[u - \widetilde{u}] = \int dx dt \left(\frac{\partial u(x,t)}{\partial x} - \frac{\partial \widetilde{u}(x,t)}{\partial x}\right)^2 + \frac{1}{2} \int dx \, (u(x,T) - \widetilde{u}(x,T))^2, \quad (19)$$

and the upper bound Repin & Tomar (2010) for approximation that exactly fulfills initial condition reads

$$U_{CD}[\widetilde{u}, f, a] = \sqrt{\int dx dt \left(w(x,t) - \frac{\partial \widetilde{u}(x,t)}{\partial x}\right)^2}$$

$$+ \frac{1}{\pi}\sqrt{\int dx dt \left(f(x) - \frac{\partial \widetilde{u}(x,t)}{\partial t} - a\frac{\partial \widetilde{u}(x,t)}{\partial x} + \frac{\partial w(x,t)}{\partial x}\right)^2}. \quad (20)$$

For upper bound (20), we have one auxiliary scalar field $w(x,t)$.

## 4.4 ELASTOPLASTICITY

Let $K_0$, $\mu$, $k_\star$, $\delta$ are positibe constants, $u$ denotes deformation vector. Given that, elastoplastic deformations are described by the following PDE

$$\partial_1\sigma_{11}(u) + \partial_2\sigma_{21}(u) + f_1 = \partial_1\sigma_{12}(u) + \partial_2\sigma_{22}(u) + f_2 = 0, \, x_1, x_2 \in \Gamma, \, u_1|_{\partial\Gamma} = u_2|_{\partial\Gamma} = 0, \quad (21)$$

where $\Gamma = (0,1)^2$, $\partial\Gamma$ is a boundary of $\Gamma$ and

$$\sigma(u) = K_0 (\partial_1 u_1 + \partial_2 u_2) I + \gamma \left(\left\|\epsilon^D(u)\right\|_F\right) \epsilon^D(u), \, \epsilon^D(u) = \epsilon(u) - \frac{1}{2} I \operatorname{tr} \epsilon(u);$$

$$\epsilon(u) = \begin{pmatrix} \partial_1 u_1 & \frac{1}{2}(\partial_1 u_2 + \partial_2 u_1) \\ \frac{1}{2}(\partial_1 u_2 + \partial_2 u_1) & \partial_2 u_2 \end{pmatrix}, \, \gamma(t) = \begin{cases} 2\mu, \, |t| \le t_0 = \frac{k_*}{2\sqrt{\mu}}; \\ (2\mu - \delta)\frac{t_0}{|t|} + \delta, \, |t| > t_0. \end{cases} \quad (22)$$

In Repin & Xanthis (1996) authors derived upper bound for natural energy norm

$$E[u-v]^2 = \int dx \left(K_0 (\operatorname{tr}\epsilon(u - v))^2 + \delta \left\|\epsilon^D(u - v)\right\|_F^2\right) \le 2C_0 \|\eta - \sigma(v)\|_{a^\star}^2, \eta \in Q_f^\star, v \in W_2^1, \quad (23)$$

where

$$\|\tau\|_{a^\star}^2 = \int dx \frac{1}{4K_0} (\operatorname{tr}\tau)^2 + \frac{1}{2\mu} \left\|\tau^D\right\|_F^2,$$

$$C_0 = 1 + \frac{2\mu_2}{\alpha_1^*}, \, \mu_2 = \frac{2\mu - \delta}{4\mu\delta}, \, \alpha_1^* = \inf_{t^\top = t} \frac{\frac{1}{4K_0} (\operatorname{tr} t)^2 + \frac{1}{2\mu} \left\|t^D\right\|_F^2}{\|t\|_F^2}, \quad (24)$$

$$Q_f^\star = \left\{\tau : \int dx \left(\sum_{ij} \tau_{ij}\epsilon(v)_{ij} - \sum_i f_i v_i\right) = 0 \forall v \in W_2^1, \tau^\top = \tau\right\}.$$

## 5 EXPERIMENTS

We use examples of PDEs described in Section 4 to study the following questions: (i) How is the accuracy of solutions obtained with Astral loss compare with the ones obtained with residual and variational losses? (ii) How cost-efficient is Astral loss in comparison with other losses? (iii) Which loss is more robust to the irregularities of underlining PDE, e.g., anisotropy of coefficients or geometric singularities? (iv) How tight in practice is the upper bound obtained with Astral loss?

This section describes the experiments we prepared to this end and the analysis of the obtained results.

### 5.1 DATASETS

To benchmark Astral loss, we consider seven PDEs described below:

1. **Isotropic diffusion**: Isotropic diffusion refers to equation (11) in $(0,1)^2$ with $\sigma(x,y) = Ia(x,y)$ with $a = 5(z - \min z)/(\max z - \min z) + 1$, $u(x,y) = \sin(\pi x)\sin(\pi y)r$ where $r, z \sim \mathcal{N}\left(0, (1-\Delta)^{-1}\right)$ with homogeneous Dirichlet conditions.

2. **Anisotropic diffusion**: Anisotropic diffusion is similarly based on equation (11) but with an anisotropy parameter $\epsilon$ introduced into the diffusion coefficient $\sigma(x,y;\epsilon) = \begin{pmatrix} 1 & 0 \\ 0 & \epsilon^2 \end{pmatrix} a(x,y)$, and into the exact solution $u(x,y) = \sin(\pi x)\sin(\pi y)r$ where $r \sim \mathcal{N}\left(0, \left(1 - \partial_x^2 - \epsilon^2\partial_y^2\right)^{-1}\right)$.

3. **Diffusion with large mixed derivative**: This PDE is built based on the same random fields as isotropic diffusion, but with different diffusion matrix $\sigma(x,y;\delta) = \begin{pmatrix} 1 & \delta \\ \delta & 1 \end{pmatrix} a(x,y)$. The case when $\delta$ is close to $1$ is especially interesting since $\det \sigma$ becomes close to $0$.

4. **Diffusion in the L-shaped domain**: Here we consider equation (11) in domain $(0,1)^2 \setminus (0.5,1)^2$, and for simplicity take $\sigma(x,y) = I$, field $f$ is sampled from $\mathcal{N}\left(0, (1 - 0.01\Delta)^{-2}\right)$ so it is smoother that for other diffusion equations.

5. **Maxwell's equation**: We consider equation (14) with $\alpha = 1$, $B = \operatorname{curl} A$, $A \sim \mathcal{N}\left(0, (1-\Delta)^{-1}\right)$ with homogeneous Neumann boundary conditions, $\mu = 5(z - \min z)/(\max z - \min z) + 1$, where $z \sim \mathcal{N}\left(0, (1-\Delta)^{-1}\right)$ with homogeneous Dirichlet conditions.

6. **Magnetostatics**: This equation is precisely the same as Maxwell's equation above but with $\alpha = 0$.

7. **Convection-diffusion equation**: This dataset is based on equation (18) with $T = 0.1$, $a$ sampled uniformly from $0$ to $10$, $f, \phi \sim \mathcal{N}\left(0, \left(1 + 0.04\left(-\partial_x^2 + a\partial_x\right)^2\right)^2\right)$ with homogeneous Dirichlet conditions.

8. **Elastoplasticity**: Dataset is based on equation (21) with $K_0 = \mu = \delta = 1$, $k_\star = 20$ and $u_1$ with $u_2$ independently drawn from $\mathcal{N}\left(0, (1-\Delta)^{-2}\right)$ with homogeneous Dirichlet conditions.

### 5.2 LOSS FUNCTIONS

For datasets based on diffusion equation, we use Astral loss function (13), standard residual loss function which can be obtained by evaluating (11) at a set of points and taking $l_2$ norm from the difference between left- and right-hand sides, and variational loss function reads

$$V[\phi] = \int dx dy \left( \frac{1}{2} \left\| \sigma^{1/2}(x,y) \operatorname{grad} \phi(x,y) \right\|^2 - \phi(x,y)f(x,y) \right). \tag{25}$$

Table 1: Results for isotropic and anisotropic diffusion equations: relative $L_2$ is relative error in $L_2$ norm, $E[\phi - \widetilde{\phi}]$ is error in natural energy norm, the large $\epsilon$ the more anisotropic is diffusion equation. Relative error is measured in $\%$ and energy norm and majorant are multiplied by $10^2$.

| | Residual | | Astral | | | Variational | |
|---|---|---|---|---|---|---|---|
| $\epsilon$ | relative $L_2$ | $E[\phi - \widetilde{\phi}]$ | relative $L_2$ | $E[\phi - \widetilde{\phi}]$ | majorant | relative $L_2$ | $E[\phi - \widetilde{\phi}]$ |
| 1 | $0.13 \pm 0.07$ | $0.03 \pm 0.01$ | $\mathbf{0.11 \pm 0.05}$ | $0.03 \pm 0.01$ | $0.13 \pm 0.03$ | $3.48 \pm 1.46$ | $1.01 \pm 0.32$ |
| 5 | $0.63 \pm 0.27$ | $0.04 \pm 0.01$ | $\mathbf{0.53 \pm 0.19}$ | $0.04 \pm 0.00$ | $0.09 \pm 0.02$ | $6.89 \pm 3.88$ | $0.62 \pm 0.22$ |
| 10 | $1.65 \pm 0.92$ | $0.07 \pm 0.03$ | $\mathbf{0.97 \pm 0.57}$ | $0.05 \pm 0.03$ | $0.11 \pm 0.03$ | $10.55 \pm 5.88$ | $0.48 \pm 0.21$ |
| 15 | $3.16 \pm 1.74$ | $0.09 \pm 0.04$ | $\mathbf{2.08 \pm 1.24}$ | $0.07 \pm 0.04$ | $0.12 \pm 0.04$ | $11.92 \pm 6.04$ | $0.49 \pm 0.19$ |
| 20 | $5.64 \pm 3.18$ | $0.12 \pm 0.05$ | $\mathbf{3.6 \pm 2.18}$ | $0.09 \pm 0.05$ | $0.13 \pm 0.06$ | $14.07 \pm 7.69$ | $0.44 \pm 0.22$ |

For Maxwell's problems, Astral losses are (16) and (17) when $\alpha = 0$. For the convection-diffusion equation, Astral loss is (20). For elastoplastic problem Astral loss is (23) with $Q_f^\star$ softly enforced with extra term. Residual losses for these problems are self-evident, and variational losses are unavailable. For magnetostatics problem (14) with $\alpha = 0$ for both residual and Astral losses, we enforce the predicted field to be solenoidal with an additional term.

### 5.3 METRICS

To evaluate training methods, we use several metrics: (i) **Relative error**. This metrics is the most often used $\sqrt{\sum_{i,j} \left( \phi_{i,j} - \widetilde{\phi}_{i,j} \right)^2} / \sqrt{\sum_{i,j} \phi_{i,j}^2}$, where $\phi_{i,j} = \phi(x_i, y_j)$ is a field computed on the uniform grid with different resolution from the training grid; (ii) **Natural energy norm**. For diffusion equation is given by equation (12), for convection-diffusion – (19), for Maxwell's equation – (15). In all cases, integrals are computed with Gauss–Legendre quadrature; (iii) **Error majorant**. The definition coincides with Astral loss, but the integral is evaluated with Gauss–Legendre quadrature on a fine grid different from the training grid. This metric is only applicable to Astral loss.

### 5.4 ARCHITECTURES AND TRAINING DETAILS

Following best practices Wang et al. (2023) we approximate all integrals with the Monte Carlo method with a small number of randomly selected points restricted to the uniform grid $64 \times 64$. Legendre grid and uniform grid used for evaluation have 200 points along each direction, for all equations but diffusion in L-shaped domain where they have 100 points in each of three square subdomains. In all cases, we enforce boundary and initial conditions exactly. For the L-shaped domain this is done with the help of mean value coordinates Floater (2003) as explained in Sukumar & Srivastava (2022). Architecture used is Siren network Sitzmann et al. (2020) with the same number of hidden neurons $N_{\text{hidden}} \in [50, 100]$ in each layer $N_{\text{layers}} \in [3, 4, 5]$. Individual Siren network is used for each field one needs to predict for a given problem. We use Lion optimizer Chen et al. (2023b) with learning rate $\in \left[ 10^{-3}, 5 \cdot 10^{-3}, 10^{-4} \right]$, exponential learning rate decay with decay rate 0.5 and transition steps $\in [10000, 25000, 50000]$. The batch size (number of randomly selected points to compute loss function) used is $16 \times 16$ and the number of weights updates is 50000. We report the average and standard deviation of all metrics of interest for 100 problems for each PDE. The best results are reported among all hyperparameters. All experiments were performed on a single Tesla V100-SXM2-16GB. Ensemble training was used, so 100 neural networks are trained simultaneously. The same setup was used to report wall-clock training time. Depending on the architecture and loss function training time is in-between 70 and 1200 seconds for 100 neural networks.

### 5.5 REPRODUCIBILITY

In `https://anonymous.4open.science/r/astral-4ECB` one can find datasets, scripts used to generate datasets, and scripts used to obtain all training results. The dependencies are mini-

Table 2: Results for convection-diffusion equation, Maxwell's equation, magnetostatics problem, diffusion with mixed derivative and elastoplasticity equation: relative $L_2$ is a relative error in $L_2$ norm, $E[\phi - \widetilde{\phi}]$ is an error in natural energy norm, the large $\delta$ the more role has a term with mixed derivatives. Relative error is measured in $\%$ and energy norm and majorant are multiplied by $10^2$.

| | Residual | | Astral | | |
|---|---|---|---|---|---|
| equation | relative $L_2$ | $E[\phi - \widetilde{\phi}]$ | relative $L_2$ | $E[\phi - \widetilde{\phi}]$ | majorant |
| conv-diff | $\mathbf{3.96 \pm 4.91}$ | $7.36 \pm 12.54$ | $4.05 \pm 4.97$ | $8.84 \pm 12.91$ | $17.91 \pm 29.76$ |
| Maxwell, $\alpha = 1$ | $5.49 \pm 2.35$ | $1.93 \pm 0.66$ | $\mathbf{0.45 \pm 0.16}$ | $0.32 \pm 0.06$ | $3.53 \pm 1.08$ |
| Maxwell, $\alpha = 0$ | $0.12 \pm 0.06$ | $0.15 \pm 0.03$ | $\mathbf{0.07 \pm 0.03}$ | $0.29 \pm 0.06$ | $0.96 \pm 0.26$ |
| L-shaped | $\mathbf{0.26 \pm 0.06}$ | $0.17 \pm 0.1$ | $0.8 \pm 0.34$ | $0.32 \pm 0.21$ | $1.29 \pm 0.75$ |
| mixed-diff, $\delta = 0.5$ | $0.13 \pm 0.07$ | $0.02 \pm 0.01$ | $\mathbf{0.12 \pm 0.06}$ | $0.05 \pm 0.01$ | $0.18 \pm 0.04$ |
| mixed-diff, $\delta = 0.7$ | $0.15 \pm 0.08$ | $0.03 \pm 0.01$ | $\mathbf{0.12 \pm 0.06}$ | $0.05 \pm 0.01$ | $0.23 \pm 0.06$ |
| mixed-diff, $\delta = 0.9$ | $0.18 \pm 0.09$ | $0.04 \pm 0.01$ | $\mathbf{0.13 \pm 0.07}$ | $0.07 \pm 0.02$ | $0.25 \pm 0.05$ |
| mixed-diff, $\delta = 0.99$ | $0.25 \pm 0.14$ | $0.15 \pm 0.05$ | $\mathbf{0.12 \pm 0.10}$ | $0.03 \pm 0.02$ | $0.76 \pm 0.16$ |
| elastoplasticity | $\mathbf{0.83 \pm 0.65}$ | $\mathbf{0.003 \pm 0.001}$ | $1.75 \pm 1.27$ | $0.007 \pm 0.001$ | $0.019 \pm 0.005$ |

mal: we use JAX, Optax DeepMind et al. (2020) and Equinox Kidger & Garcia (2021). Scripts with dataset collection also use NumPy Harris et al. (2020) and SciPy Virtanen et al. (2020).

## 5.6 RESULTS

Results are summarized in Table 1 and Table 2; training wall clock time is reported in Table 3; error distribution and learning curves for selected equations are available in Appendix B. The main observations are:

1. **Accuracy**. When ranked concerning the $L_2$ norm we can see that residual and Astral losses are much better than variational loss. On most of the problems, Astral loss leads to similar or slightly better error than residual loss. There are two notable exceptions: for Maxwell's equations Astral leads to a much better error and for the L-shaped domain the error reached by residual loss is better.

2. **Robustness**. When problems become less regular Astral loss leads to better results than residual loss. This is the case for both highly anisotropic diffusion equations and diffusion equations with large mixed derivatives. Also note that Astral upper bound does not deteriorate for diffusion equation in the L-shaped domain which is a standard test problem used to benchmark methods for a posteriori error estimation.

3. **Cost-efficiency**. From Table 3 we can see that even though training with Astral requires predicting more fields, the training time is typically smaller than for residual loss. The reason for that is one does not need to compute second derivatives. It is often the case that Astral loss also requires smaller networks for best error, e.g., for Maxwell's equation (Table 2, $\alpha = 1$) training time for Astral loss is 105 seconds and for residual loss is 1176 seconds, yet Astral loss leads to an order of magnitude better relative error.

4. **Quality of the upper bound**. Here we observe that the upper bound is typically reasonably tight. In the most favorable cases, one has mild overestimation by a factor of 0.5 (anisotropic equation $\epsilon = 5$). For some cases, overestimation is more severe, e.g., by a factor of 2.0 for the convection-diffusion problem, and by a factor of 10 for Maxwell's equation. But overall upper bound remains a good estimate of the error in energy norm. It is also possible to build a statistical model that predict error if a dataset with exact solutions is available. This possibility is investigated in Appendix E where we show that the value of Astral loss is much better predictor for error than the value of the residual loss.

## 6 CONCLUSION

We introduced a novel type of loss that allows for error estimation. The advantages of the loss include: (i) When Astral is used, one can compute the upper bound on error (compute integral

Table 3: Average training time in seconds required for 50000 updates of 100 neural network weights on a single Tesla V100-SXM2-16GB. The small neural network has 50 hidden neurons per layer and 3 hidden layers, large neural network has 100 hidden neurons per layer and 5 hidden layers.

| equation | Residual | | Astral | | Variational | |
|---|---|---|---|---|---|---|
| | small | large | small | large | small | large |
| diff | 128 | 535 | 92 | 437 | 37 | 164 |
| conv-diff | 77 | 326 | 62 | 276 | – | – |
| Maxwell, $\alpha = 1$ | 298 | 1176 | 105 | 481 | – | – |
| Maxwell, $\alpha = 0$ | 338 | 1312 | 142 | 598 | – | – |
| L-shaped | 133 | 497 | 110 | 480 | – | – |
| mixed-diff | 237 | 728 | 91 | 428 | – | – |

alike (35)) and use it for quality control. This is not possible with other losses; (ii) For second-order problems Astral loss requires only the first derivative, which speeds up training compared to residual loss; (iii) For most equations, Astral loss leads to better final relative error.

Limitations are as follows: (i) One needs to derive the upper bound. Although upper bounds are available for many relevant equations, it is challenging to derive them for arbitrary PDE. It is important to note that this is also the case for residual loss, defined only for PDEs when a strong form makes sense; (ii) Integrals involved in the upper bound can be hard to evaluate reliably. The problem is that the error term is not straightforward to control when a neural network is used as an ansatz, and one can observe overfitting Rivera et al. (2022). This can result in cases when, because of numerical errors, the upper bound is smaller than the error, although this is theoretically impossible.

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

# A   ASTRAL LOSS FOR SIMPLE BOUNDARY VALUE PROBLEM

Consider a generalisation of boundary-value problem (1)[1]

$$-\frac{d}{dx}\left(a(x)\frac{d\phi(x)}{dx}\right) = f(x), \; x \in (-1, 1), \; \phi(-1) = \phi(1) = 0, \; a(x) \geq \epsilon > 0. \qquad (26)$$

Following Mali et al. (2013), we will estimate deviation from the exact solution in a natural energy norm

$$E[\phi - \widetilde{\phi}] = \sqrt{\int dx\, a(x) \left(\frac{d\phi(x)}{dx} - \frac{d\widetilde{\phi}(x)}{dx}\right)^2}. \qquad (27)$$

Note that since

$$\int_{-1}^{1} dx\, a(x)\frac{du(x)}{dx} \geq \lambda_{\min} \int_{-1}^{1} dx\, u^2(x) =: \lambda_{\min} \|u\|_2^2, \qquad (28)$$

where $\lambda_{\min}$ is a minimal eigenvalue of the differential operator (26), the error in energy norm is an upper bound for the error in $L_2$ norm

$$\left\|\phi - \widetilde{\phi}\right\|_2 \leq \frac{1}{\sqrt{\lambda_{\min}}} E[\phi - \widetilde{\phi}]. \qquad (29)$$

The first step is to subtract the approximate solution from both sides of the weak form of (26)

$$\int_{-1}^{1} dx\, \frac{dw(x)}{dx} a(x) \left(\frac{d\phi(x)}{dx} - \frac{d\widetilde{\phi}(x)}{dx}\right) = \int dx\, \left(w(x)f(x) - a(x)\frac{dw(x)}{dx}\frac{d\widetilde{\phi}}{dx}\right). \qquad (30)$$

If we take $w(x) = \phi(x) - \widetilde{\phi}(x)$ in the expression above we will have $E[\phi - \widetilde{\phi}]^2$ on the left. Our strategy is to bound the right-hand side from above with the expression proportional to $E[\phi - \widetilde{\phi}]$, with the remaining factor free from unknown exact solution $\phi$. To do that we introduce two extra terms using the following identity ($w(\pm 1) = 0$)

$$\int_{-1}^{1} dx\, \left(\frac{dw(x)}{dx} y(x) + w(x)\frac{dy(x)}{dx}\right) = 0, \qquad (31)$$

where $y(x)$ is arbitrary function. With these two terms right-hand side of (30) reads

$$\int dx\, \left(w(x)\left(f(x) - \frac{dy(x)}{dx}\right) + \frac{dw(x)}{dx}\left(-a(x)\frac{d\widetilde{\phi}}{dx} + y(x)\right)\right). \qquad (32)$$

Now, we take $w(x) = \phi(x) - \widetilde{\phi}(x)$ and from Cauchy–Schwarz inequality obtain the bound on the first term of (32)

$$\int_{-1}^{1} dx\, \left(w(x)\left(f(x) - \frac{dy(x)}{dx}\right)\right) \leq \left\|\phi - \widetilde{\phi}\right\|_2 \left\|f - \frac{dy}{dx}\right\|_2 \leq \frac{1}{\sqrt{\lambda_{\min}}} E\left[\phi - \widetilde{\phi}\right]\left\|f - \frac{dy}{dx}\right\|_2, \qquad (33)$$

where $\|g\|_2 = \sqrt{\int dx\, g^2(x)}$.

For the second term in (32) we similarly obtain

$$\int_{-1}^{1} dx\, a^{1/2}(x)\frac{dw(x)}{dx}\frac{1}{a^{1/2}(x)}\left(-a(x)\frac{d\widetilde{\phi}(x)}{dx} + y(x)\right) \leq E\left[\phi - \widetilde{\phi}\right]\left\|\frac{1}{a^{1/2}}\left(y - a\frac{d\widetilde{\phi}}{dx}\right)\right\|_2, \qquad (34)$$

where we introduced $a^{1/2}$ in the numerator and denominator and made use of Cauchy–Schwarz inequality.

---

[1]For simplicity we omit technical, see Mali et al. (2013). For this particular problem $f$ belongs to $L^2$, approximate solution belongs to Sobolev space $H^1$ and is supposed to agree exactly with boundary conditions, trial functions are from $H^1$ and zero on the boundary, auxiliarry function is also from $H^1$.

These two bounds combined give us an upper bound on the error in energy norm after we cancel $E[\phi - \widetilde{\phi}]$ from both sides:

$$E[\phi - \widetilde{\phi}] \leq \frac{1}{\sqrt{\lambda_{\min}}} \left\| f - \frac{dy}{dx} \right\|_2 + \left\| \frac{1}{a^{1/2}} \left( y - a\frac{d\widetilde{\phi}}{dx} \right) \right\|_2. \tag{35}$$

Since the upper bound contains no unknown parameters, one can parametrize $\widetilde{\phi}$ and $y$ with a neural network and use the right-hand side of (35) as a loss function in a physics-informed setting. In this paper, we call **Astral** losses alike the right-hand side of (35) based on error majorants.

Is it a good loss function? We can look at the point of the requirements listed above: (i) The bond is tight. Observe that if $y = a\frac{d\widetilde{\phi}}{dx}$ and $\widetilde{\phi} = \phi$ the bound is saturated; (ii) The bound does not contain $\phi$, it can be computed by either optimization of $y$ for fixed $\widetilde{\phi}$ or by joint optimization of $\widetilde{\phi}$ and $y$; (iii) The bound appears as a continuous functional that does not depend on a particular form of approximation made, so it remains valid for finite-differences, finite-elements, physics-informed neural networks and all other discretization methods; (iv) To compute the upper bound one needs to find first derivatives of two fields $\widetilde{\phi}$ and $y$, whereas for the residual loss one need to find second and first derivative of $\phi$. One can expect that numerical costs are comparable, which we will verify in Section 5.

The derivation of Astral loss for simple BVP relies on straightforward inequalities. More elaborate techniques suitable for more complex equations can be found in Mali et al. (2013) and references therein.

# B ERROR DISTRIBUTIONS AND LEARNING CURVES

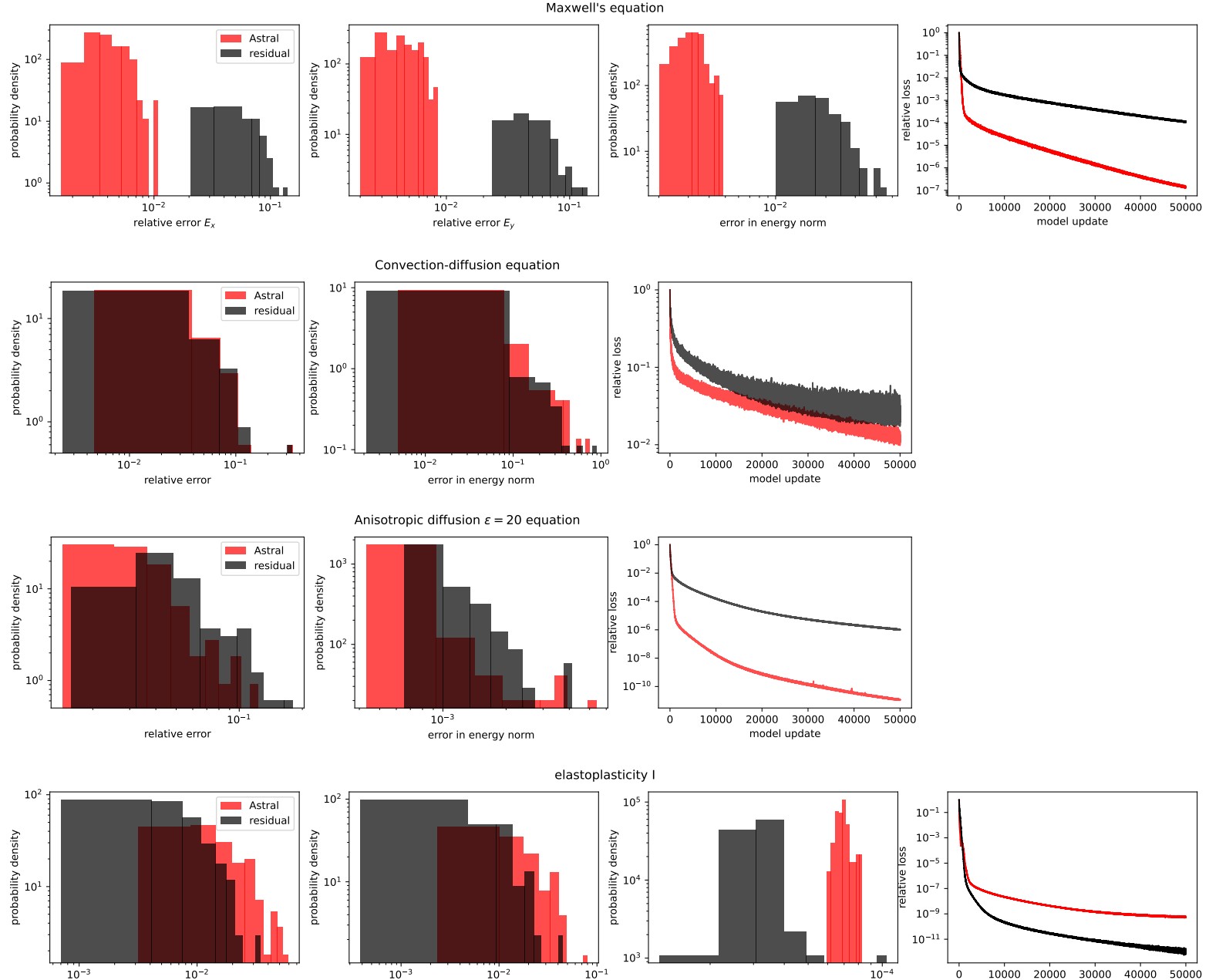

## C  SPATIAL DISTIRBUTION OF ERROR, RESIDUAL, DENSITY OF ENERGY NORM

Comparison of spatial distribution of residual, error and density of error energy norm for several stationary diffusion equation with different diffusion coefficients. Red dot corresponds to the point with maximal residual. Residual is poorly correlated with both energy and density of error energy norm. Average spatial correlation coefficient over 100 randomly selected PDEs is $0.22 \pm 0.09$.

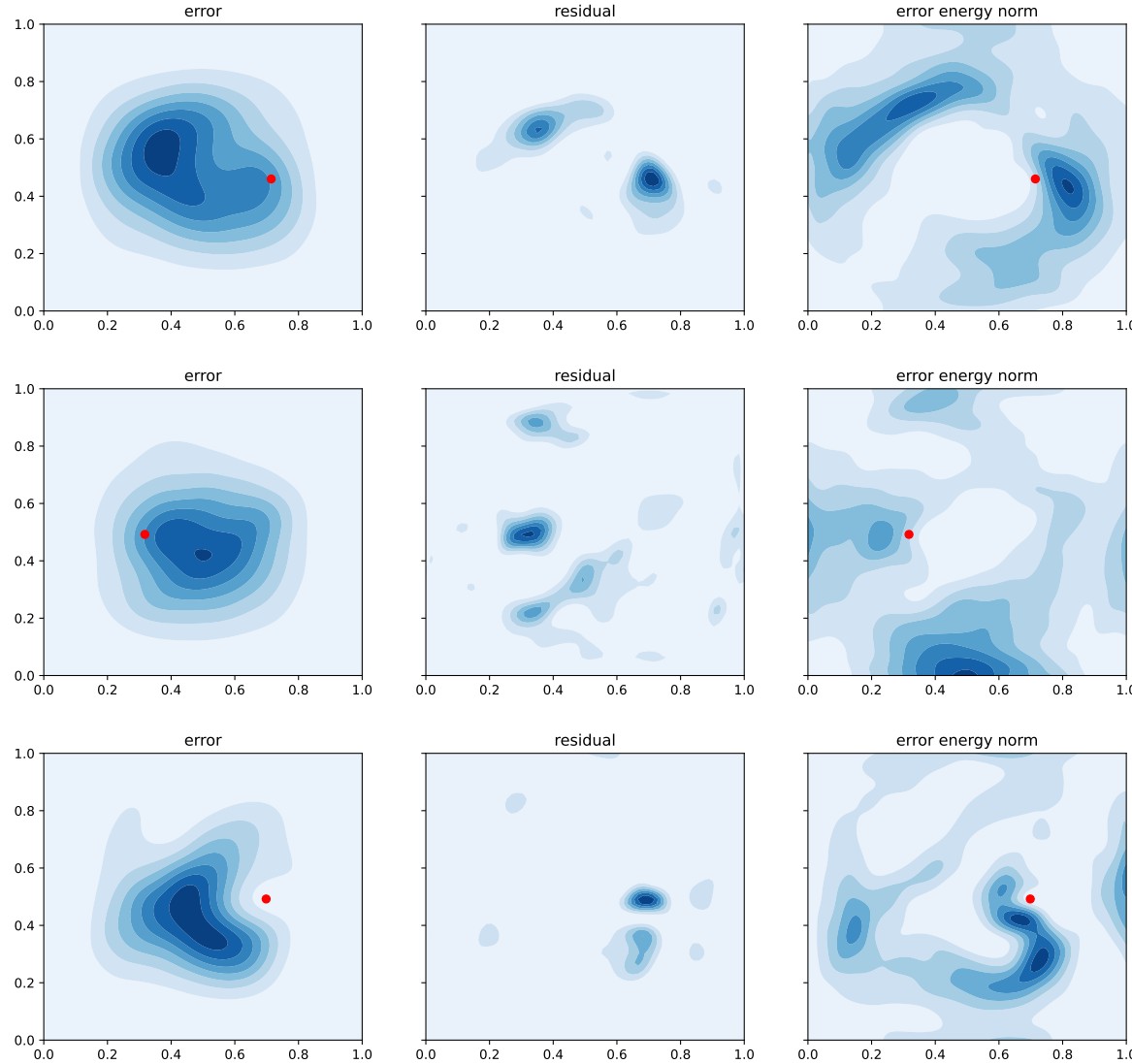

# D SPATIAL DISTIRBUTION OF ERROR, ERROR INDICATOR, DENSITY OF ENERGY NORM

Comparison of spatial distribution of error indicator, error and density of error energy norm for several stationary diffusion equation with different diffusion coefficients. Error indicator is poorly correlated with energy but well correlated with density of error energy norm. Average spatial correlation coefficient over 100 randomly selected PDEs is $0.82 \pm 0.04$.

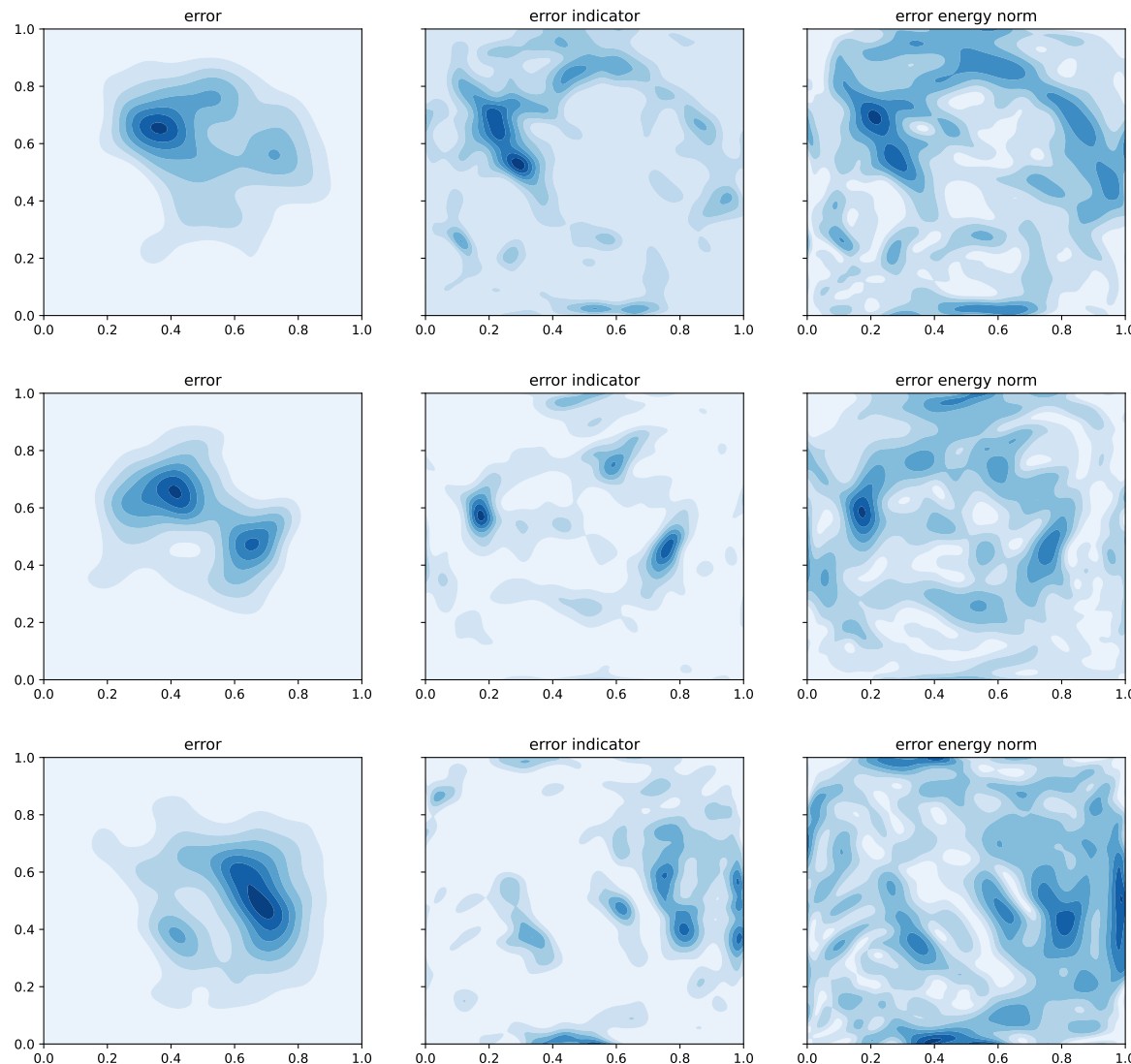

# E    STATISTICAL CORRELATION: ERROR VS RESIDUAL, ENERGY NORM VS RESIDUAL AND ASTRAL LOSS

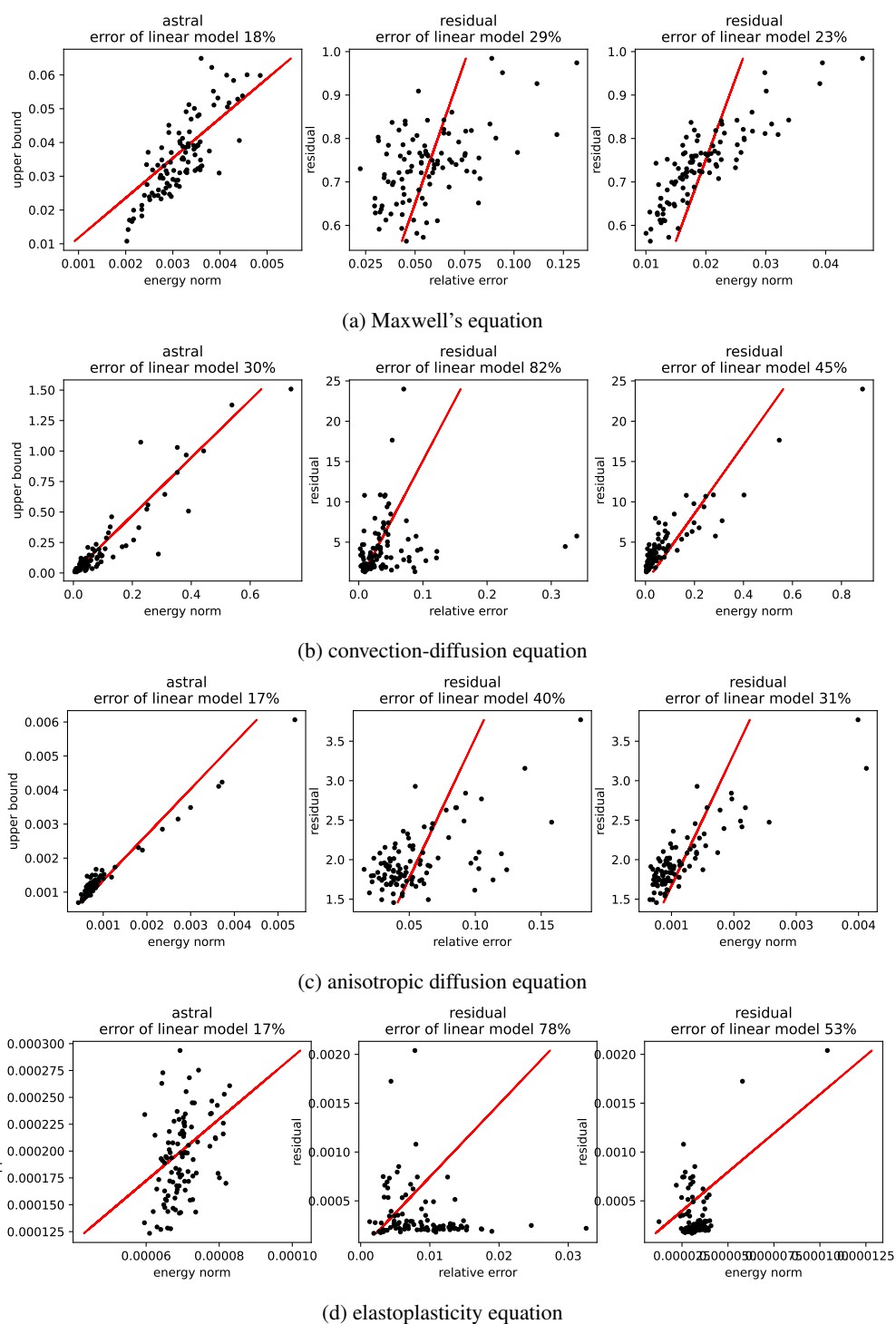

(a) Maxwell's equation

(b) convection-diffusion equation

(c) anisotropic diffusion equation

(d) elastoplasticity equation

A posteriori error estimate allows to obtain error for each individual approximate solution without knowing the exact solution. The other possibility is to use dataset with known solutions to build a statistical model that predict error for a given method. Results of such experiment are available in

the figure above. Evidently, value of Astral loss provide a much better feature for linear model than the value of residual.

