# OpenReview forum: "Astral: training physics-informed neural networks with error majorants"
_ICLR.cc/2025/Conference — Submitted to ICLR 2025_

### Official Review · Reviewer_PvTH · 2024-10-17

**Soundness:** 3
**Presentation:** 3
**Contribution:** 4
**Rating:** 6
**Confidence:** 4

**Summary:**

The paper proposes training physics-informed neural networks (PINNs) not via the usual residual-based loss, but via minimizing the so-called error majorant, a quantity that can be derived for certain classes of PDEs but that requires learning surrogate functions. The paper shows that the energy norm error correlates with the relative error of the PINN candidate function to a better degree than the residuals do. The results indicate that this approach of training PINNs results in smaller relative errors in many cases, and that further the runtime to achieve this error is smaller than for conventional training strategies.

**Strengths:**

The idea of training PINNs with the error majorant is a novel idea that, at least for some PDE classes, appears to be quite promising. The paper is generally well written and, for the most part, quite accessible. I particularly commend the intuitive motivation in Section 2.1, and the fact that a large number of PINNs (100) were trained for each setting, thus ruling out random effects. Some readers may consider the fact that the majorant must be derived/available for the considered PDE class as a shortcoming. In my experience with PINNs, in most forward problems, training often requires problem-specific adaptations (such as domain decomposition for large domains, collocation point weighting for stiff problems, etc.). I thus see the proposed approach as yet another such problem-specific adaption, but as an innovative one that leaves the established PINN paradigm of using the residual loss.

**Weaknesses:**

At some parts, the paper is not well written and somewhat unclear. For example, I think the authors switch between $\phi$ in Section 3, which is a general notation for the solution of the PDE, and problem-specific notation ($\phi$, $B$, $u$, etc.) in Section 4. In Section 5, the solution for diffusion is given as $u$, while in Section 4 it is given as $\phi$ (if I understand correctly). In Section 2.3, the surrogate function (flux) is denoted as $\tilde{F}$, while I think this corresponds to $\omega$ in Section 3. This makes reading the paper quite taxing. More importantly, it is sometimes not clear which functions are given and which are learned; and of those that are learned, which are the solution to the PDE ($\phi$) and which are surrogate functions ($\omega$). I thus suggest to homogenize notation.

Section 4 could be moved to the appendix, and the example in Section 2.3 or the general framework in Section 3 could be expanded. I think it must be more clear that surrogate functions need to be learned, which may be best accomplished by a schematic.

There are further considerations regarding the experimental section:
- In Section 5, it is not clear how the variational approach works. A few lines explaining this should be included in the revision.
- To judge the difficulty of the problems, I suggest to plot the solutions and PINN candidates for each of the considered problems.
- Table 1 could use vertical bars between Residual / Astral / Variational to better separate the results.
- Table 2 shows that the majorant is often a quite loose bound, contradicting the claims of the paper somewhat. Indeed, also for the mixed diffusion equation, the majorant is at least a factor of 3-4 of the natural energy error. It would be valuable to explain this phenomenon, and maybe also illustrate the error of the surrogate function required to approximate the natural energy error (e.g., the flux).
- Figure 1 requires axis labels.

Finally, I suggest that the paper undergoes proof reading by a native speaker, as some constructs read strangely and do not parse well.

**Questions:**

- What is the surrogate function $\omega$ for the elastoplasticity example in Section 4.4?
- How exactly does the variational approach work?
- What do the surrogate functions required for evaluating the natural energy error look like? What is the behavior of their errors?

---

> ### Author Response · Authors · 2024-11-25
>
> ```
> In Section 5, it is not clear how the variational approach works. A few lines explaining this should be included in the revision.
> ```
> also
> ```
> How exactly does the variational approach work?
> ```
> For this part we provided clear reference. Variational method is well known and hardly needed any introduction. Interested reader can consult either the original publication or the attached code where this method is implemented.
>
> ```
> To judge the difficulty of the problems, I suggest to plot the solutions and PINN candidates for each of the considered problems.
> ```
> Most difficulties for PiNN arise from the residual formualtion not a complexity of the solution. This is especially evident since siren networks and Fourier Features are also used for NeRF which can fit extremely complicated shapes.
>
> ```
> Table 2 shows that the majorant is often a quite loose bound, contradicting the claims of the paper somewhat. Indeed, also for the mixed diffusion equation, the majorant is at least a factor of 3-4 of the natural energy error. It would be valuable to explain this phenomenon, and maybe also illustrate the error of the surrogate function required to approximate the natural energy error (e.g., the flux).
> ```
> Factor $3-4$ is a SOTA for a posteriori error estimate. One can hardly hope for an improvement over these numbers.
> ```
> Figure 1 requires axis labels.
> ```
> Figure 1 compare residual, error, error indicator, error in energy norm. All coordinates are in the intervals $[0, 1]$, which is hardly relevant for visually comparing spatial distributions.
>
> ```
> What is the surrogate function $\omega$ for the elastoplasticity example in Section 4.4?
> ```
> and
> ```
> More importantly, it is sometimes not clear which functions are given and which are learned; and of those that are learned, which are the solution to the PDE ($\phi$) and which are surrogate functions ($\omega$).
> ```
> Word "surrogate" is never used in our paper. The auxiliary functions for elastoplasitcity example are all function not available in the formulation of the problem. The same goes for all other problems. If function is not available in the formulation of PDE, this is auxiliary function.
> ```
> What do the surrogate functions required for evaluating the natural energy error look like? What is the behavior of their errors?
> ```
> This is, unfortunately, an ill-posed question. Auxiliary functions will depend on the approximate solution and the best one is unknown and can be not computable in the selected approximation space. These function often have "physical meaning" which is apparent from the formulation of the problem. For example, in elliptic problem auxiliary functions corresponds to fluxes of the solution.

---

> > ### Comment · Reviewer_PvTH · 2024-11-27
> >
> > Thanks for the reply. I will keep my score.

---

### Official Review · Reviewer_gM3w · 2024-11-02

**Soundness:** 3
**Presentation:** 3
**Contribution:** 2
**Rating:** 5
**Confidence:** 4

**Summary:**

The manuscript proposes alternatives to the common, residual-based loss functions used in physics-informed neural networks. The authors argue that certain functional a posteriori error estimators better match the error (measured in energy norm) than the standard PINN loss formulation. Numerical examples for a number of different equations are presented comparing the standard residual/PINN approach to the one using a posteriori error estimators.

**Strengths:**

- Moving away from strong formulations to first-order systems (the error majorants presented are not exactly first order systems, but relatively close) seems like a good idea.

- Good error estimators are useful in practice.

**Weaknesses:**

- Novelty: Although the proposed loss functions are not exactly first-order system reformulations of the considered PDEs, they share a similar spirit -- no second derivatives are needed but instead auxiliary variables are introduced. However, first-order system formulations are not novel, not even for neural network based solution methods for PDEs, see for example the works of [Cai, arXiv:1911.02109] or [Schwab, arXiv:2409.20264]. So I believe it is crucial to understand if the advantages of the proposed loss functions are due to the reformulation as a first-order system or if they are specific to the proposed losses. Can the authors comment on that? The advantages of first-order systems that I have in mind are: better conditioning than PINN type losses (thinking in terms of FEM results), and the fact that only first derivatives need to be computed.

- Numerical results: The presented results confirm an improvement over the PINN baseline (in terms of errors), but it is not drastic. The relative L2 errors even seem to be comparable for the majority of the considered equations. Thus the value of the error majorants may lie mostly in what they are designed for -- estimating the error (after training) and the real question is: Are first-order systems (or Astral loss) computationally more efficient than the standard PINN formulation. This question is not sufficiently addressed in the present manuscript, and would need the experiments to focus on runtime and a thorough evaluation of automatic differentiation and implementation tricks. (For instance, it is often more efficient to compute the spatial derivatives in forward mode, incorporate tricks like the Forward Laplacian framework [ arXiv:2307.08214]  etc.)

- Recent work shows that PINNs are best optimized using second order methods. Improvement in accuracy can be drastic when changing from a stochastic first order method to a natural gradient or Gauss-Newton method see [Müller, arXiv:2302.13163] and [Rathore, arXiv:2402.01868]. It is important to take these recent developments into consideration for a thorough evaluation.

**Questions:**

- More of a remark than a question: In Section 2.1, the authors give two motivating examples. The first one (the sinusoidal solution) shows that the function can be L2 close to the solution while having a large residual. The authors use this as an argument against residual based loss functions. I don't think this is fair, the same phenomenon happens for Astral loss. Can the authors comment?

- The standard PINN residual is, in fact, also an a posteriori error estimator for the error. Even for the error in the regularity spaces (H2 for Poisson etc). See the results in [Zeinhofer, arXiv:2311.00529]. Can the authors discuss why they prefer different a posteriori error estimators? I suppose it is because of unknown constants, but I would appreciate a clarification.

---

> ### Author Response · Authors · 2024-11-25
>
> ```
> Novelty: Although the proposed loss functions are not exactly first-order system reformulations of the considered PDEs, they share a similar spirit -- no second derivatives are needed but instead auxiliary variables are introduced. However, first-order system formulations are not novel, not even for neural network based solution methods for PDEs, see for example the works of [Cai, arXiv:1911.02109] or [Schwab, arXiv:2409.20264]. So I believe it is crucial to understand if the advantages of the proposed loss functions are due to the reformulation as a first-order system or if they are specific to the proposed losses. Can the authors comment on that? The advantages of first-order systems that I have in mind are: better conditioning than PINN type losses (thinking in terms of FEM results), and the fact that only first derivatives need to be computed.
> ```
> We do not claim to be the first to introduce the first-order formulation. In Section 2.3 we made an analogy with MIMO which also uses firs-order formulation. The work of Cai (arXiv:1911.02109) is discussed in the Introduction. The recent work of Schwab (arXiv:2409.20264) looks interesting. However, it was not available to us at the time of submission (submission deadline -- October 1, paper appears on arxiv -- September 30).
>
> While our method offers comparable accuracy, our primary contribution lies in the rigorous analysis of the error.
>
> ```
> Numerical results: The presented results confirm an improvement over the PINN baseline (in terms of errors), but it is not drastic. The relative L2 errors even seem to be comparable for the majority of the considered equations. Thus the value of the error majorants may lie mostly in what they are designed for -- estimating the error (after training) and the real question is: Are first-order systems (or Astral loss) computationally more efficient than the standard PINN formulation. This question is not sufficiently addressed in the present manuscript, and would need the experiments to focus on runtime and a thorough evaluation of automatic differentiation and implementation tricks. (For instance, it is often more efficient to compute the spatial derivatives in forward mode, incorporate tricks like the Forward Laplacian framework [ arXiv:2307.08214] etc.)
> ```
> This can be done, but is largely besides the point. The end of time comparison was to show that training additional fields is not going to compromise the performance significantly. Besides, any tricks applicable to standard loss will also speed up Astral loss as well.
>
> ```
> Recent work shows that PINNs are best optimized using second order methods. Improvement in accuracy can be drastic when changing from a stochastic first order method to a natural gradient or Gauss-Newton method see [Müller, arXiv:2302.13163] and [Rathore, arXiv:2402.01868]. It is important to take these recent developments into consideration for a thorough evaluation.
> ```
> Again, any improvement in optimisation can be applied regardless of the loss function.
>
> ```
> More of a remark than a question: In Section 2.1, the authors give two motivating examples. The first one (the sinusoidal solution) shows that the function can be L2 close to the solution while having a large residual. The authors use this as an argument against residual based loss functions. I don't think this is fair, the same phenomenon happens for Astral loss. Can the authors comment?
> ```
> This does not happen for Astral loss since it contains derivatives of the solution, so in energy norm this perturbation is large.
>
> ```
> The standard PINN residual is, in fact, also an a posteriori error estimator for the error. Even for the error in the regularity spaces (H2 for Poisson etc). See the results in [Zeinhofer, arXiv:2311.00529]. Can the authors discuss why they prefer different a posteriori error estimators? I suppose it is because of unknown constants, but I would appreciate a clarification.
> ```
> We agree that a suggested work contain a result that is a special case of our framework. We prefer different a posteriori error estimate because functional a posteriori error estimate is applicable to wider set of equations.

---

> > ### Comment · Reviewer_gM3w · 2024-11-25
> >
> > Thanks for the reply.
> >
> > First-order system vs Astral: I might have missed a point. Training with Astral is as costly as training a first-order system formulation but gives an a posteriori error estimator for free, which is better than the residual error estimator (of the strong form).
> >
> > To make your work stronger, I suggest to clearly state that the residual formulation (vanilla PINN) and first order system formulations are also a posteriori error estimators for a large class of equations, but of lower quality. Furthermore, you should also compare to the a posteriori error estimators that a first-order reformulation yields.
> >
> > I don't think including recent developments in PINN optimization is besides the point, as it is integral that these advances play nicely with your proposed loss functions. But I understand it is outside the scope of the rebuttal.
> >
> > Finally back to your sinusoidal example in the beginning. I think it should be removed as it is misleading. You should neither expect the residual nor the energy of a sequence of function that merely converges in L2 to converge to the energy/resiudal of the limit of said function. In other words neither the energy nor the residual is continuous wrt to the L2 topology. You cannot use this example as a "problem" of the residual formulation.

---

### Official Review · Reviewer_8fDV · 2024-11-04

**Soundness:** 2
**Presentation:** 4
**Contribution:** 2
**Rating:** 5
**Confidence:** 3

**Summary:**

This paper starts from the observation that the loss minimization used for pinns does not generally guarantee that the distance L2 of the approximated solution from the true solution is minimized. To overcome this problem, the authors propose new losses for PINNs, which are shown to bound the L2 distance. The authors then provide empirical evidence that these losses achieve lower relative L2 errors, particularly as the equation becomes more physically complicated.

**Strengths:**

- Bounding of new losses are clearly and consistently derived.
- Clear and pedagogical presentation.

**Weaknesses:**

- The proposed solutions are specific to each equation, which limits the applicative scope of the results.
- Empirical results are not sufficiently convincing. Only a few examples show an improvement in errors, but in most cases this remains of the same order of magnitude. It is therefore unclear whether these improvements are significant, or simply the result of statistical or numerical fluctuations.
- As has already been shown in several series of works, what affects the lack of convergence of PINNs is the poor conditioning of the problem (see for example https://doi.org/10.1016/j.jcp.2021.110768 or https://arxiv.org/abs/2310.05801) and solutions have since been successfully proposed to correct this problem (see for example https://arxiv.org/abs/2302.13163 or https://arxiv.org/abs/2402.10680), with a significant impact both on the minimization of the classical loss, and on the minimization of L2 and even H1 errors.
- Finally, questions of generalizing PINNs to different losses have already been explored in detail by Siddhartha Mishra's students. We refer you, for example, to the dissertations by Tim De Ryck (https://www.research-collection.ethz.ch/bitstream/handle/20.500.11850/674112/dissertation_deryck.pdf?sequence=1) or Roberto Molinaro (https://www.research-collection.ethz.ch/bitstream/handle/20.500.11850/646749/Thesis%2813%29.pdf?sequence=1).
- At the beginning of section 2, a proper definition of error and residual would be suited.

**Questions:**

- Perhaps the improvements you've seen in your experiences come from improved conditioning of the problem. If so, this could be an interesting line of research, since as things stand, the proposed corrections require expensive computational corrections. So it would be interesting if you could provide an analysis of the NTK spectrum with classical loss and with your loss, following the work presented in https://doi.org/10.1016/j.jcp.2021.110768.

---

> ### Author Response · Authors · 2024-11-25
>
> ```
> The proposed solutions are specific to each equation, which limits the applicative scope of the results.
> ```
> We agree with your point about the limited scope of the technique, and we explicitly acknowledge this in the Conclusion. However, we don't believe this is necessarily a major flaw for our research.
>
> As an example of a similarly "limited" technique, one can consider the Deep Ritz method (https://arxiv.org/abs/1710.00211). While Deep Ritz can only be applied to PDEs with variational formulation, it's still considered valuable by the community, having been referenced in over $1000$ works.
>
> Our method also can be applied to arbitrary variational problems. This is because, in such cases, an error majorant can be constructed using the dual equation.
>
> ```
> Empirical results are not sufficiently convincing. Only a few examples show an improvement in errors, but in most cases this remains of the same order of magnitude. It is therefore unclear whether these improvements are significant, or simply the result of statistical or numerical fluctuations.
> ```
>
> We acknowledge the reviewer's point about accuracy, which is explicitly addressed in Section 5.6. However, we believe our contribution goes beyond achieving similar accuracy as standard methods. The key feature of Astral loss is its ability to provide direct control over error, both in magnitude and spatial distribution.
>
> Perhaps the reviewer's focus on accuracy overshadows the significance of a posteriori error estimation. As highlighted in Figure 1, Table 1, 2, Appendices B, C, D, and E, Astral loss offers valuable features for error control and estimation compared to residual loss. These include the quality of the upper bound, spatial correlation between error and energy norm, and statistical correlations.
>
> While accuracy is important, a broader evaluation might consider the value of tools like error estimation for specific applications. If the reviewer prioritizes accuracy, we would appreciate comments on the state-of-the-art error (around $16\%$) of PiNNs on lid-driven cavity flow (https://arxiv.org/abs/2308.08468). Should we shut down all PiNN research based on the fact that classical methods effortlessly reach $\simeq 10^{-15}$ relative error on this trivial test task?
>
> ```
> As has already been shown in several series of works, what affects the lack of convergence of PINNs is the poor conditioning of the problem (see for example https://doi.org/10.1016/j.jcp.2021.110768 or https://arxiv.org/abs/2310.05801) and solutions have since been successfully proposed to correct this problem (see for example https://arxiv.org/abs/2302.13163 or https://arxiv.org/abs/2402.10680), with a significant impact both on the minimization of the classical loss, and on the minimization of L2 and even H1 errors.
> ```
> It is certainly an interesting line of work, but we find it hardly related to our research, which is on the a posteriori error estimate.
>
> ```
> Finally, questions of generalizing PINNs to different losses have already been explored in detail by Siddhartha Mishra's students. We refer you, for example, to the dissertations by Tim De Ryck (https://www.research-collection.ethz.ch/bitstream/handle/20.500.11850/674112/dissertation_deryck.pdf?sequence=1) or Roberto Molinaro (https://www.research-collection.ethz.ch/bitstream/handle/20.500.11850/646749/Thesis%2813%29.pdf?sequence=1).
> ```
> We respectfully disagree with the reviewer's assessment. While the provided documents are undoubtedly comprehensive, we believe they are not directly relevant to the specific focus of our research, which is on a posteriori error estimation. We would appreciate more specific feedback related to our paper's contributions in this area.
>
> ```
> Perhaps the improvements you've seen in your experiences come from improved conditioning of the problem. If so, this could be an interesting line of research, since as things stand, the proposed corrections require expensive computational corrections. So it would be interesting if you could provide an analysis of the NTK spectrum with classical loss and with your loss, following the work presented in https://doi.org/10.1016/j.jcp.2021.110768.
> ```
>
> Firstly, as Table 3 demonstrates, our methods are faster than those based on residuals, negating the need for "expensive computational corrections." Secondly, NTK analysis is irrelevant to our research, which focuses on a posteriori error estimation, not training dynamics.

---

> ### Author Response · Authors · 2024-11-30
>
> Dear 8fDV,
>
> We kindly ask you to participate in the discussion. Can you please address our rebuttal? We are especially interested in the parts on accuracy and numerical efficiency.

---

> > ### Comment · Reviewer_8fDV · 2024-12-02
> >
> > Sorry for the late answer.
> > > We agree with your point about the limited scope of the technique, and we explicitly acknowledge this in the Conclusion. However, we don't believe this is necessarily a major flaw for our research.
> > > As an example of a similarly "limited" technique, one can consider the Deep Ritz method (https://arxiv.org/abs/1710.00211). While Deep Ritz can only be applied to PDEs with variational formulation, it's still considered valuable by the community, having been referenced in over works.
> > > Our method also can be applied to arbitrary variational problems. This is because, in such cases, an error majorant can be constructed using the dual equation.
> >
> > My point was just to point out that this makes the technique hard to handle for application purposes (still with the focus on accuracy). Nevertheless I acknowledge now the fact that:
> > 1. Accuracy is not your primary goal.
> > 2. Your method can be applied can be applied to any variational problem, which makes of course a relevant enough scope.
> >
> > > We acknowledge the reviewer's point about accuracy, which is explicitly addressed in Section 5.6. However, we believe our contribution goes beyond achieving similar accuracy as standard methods. The key feature of Astral loss is its ability to provide direct control over error, both in magnitude and spatial distribution.
> > > Perhaps the reviewer's focus on accuracy overshadows the significance of a posteriori error estimation. As highlighted in Figure 1, Table 1, 2, Appendices B, C, D, and E, Astral loss offers valuable features for error control and estimation compared to residual loss. These include the quality of the upper bound, spatial correlation between error and energy norm, and statistical correlations.
> >
> > I acknowledge that I probably underestimated this part, and that your work is valuable in this respect. I will reconsider my score accordingly.
> >
> > > While accuracy is important, a broader evaluation might consider the value of tools like error estimation for specific applications. If the reviewer prioritizes accuracy, we would appreciate comments on the state-of-the-art error (around 16) of PiNNs on lid-driven cavity flow (https://arxiv.org/abs/2308.08468). Should we shut down all PiNN research based on the fact that classical methods effortlessly reach $\simeq10^{-15}$ relative error on this trivial test task?
> >
> > I strongly disagree with this remark. My point was that the Natural gradient methods that I mentioned precisely lower the gap by many order of magnitudes, between what were SotA results obtained with PINN and SotA results obtained with other methods. One may for instance refer to https://arxiv.org/abs/2402.10680 to observe how Natural gradient methods can strongly lower the error obtained for fluid dynamics equation.
> > Shutting down all PiNN research is beside the point. However you should in my opinion at least discuss the interplay between your work and SotA Natural Gradient optimization methods. As mentioned by reviewer gM3w, this could even interact nicely.
> >
> > > It is certainly an interesting line of work, but we find it hardly related to our research, which is on the a posteriori error estimate.
> > Same remark : If you claim, as you do, that your work do improve the accuracy of results, you cannot ignore those work. This is independent of a posteriori estimate.
> >
> > > We respectfully disagree with the reviewer's assessment. While the provided documents are undoubtedly comprehensive, we believe they are not directly relevant to the specific focus of our research, which is on a posteriori error estimation. We would appreciate more specific feedback related to our paper's contributions in this area.
> > I am not an expert in a posteriori error estimation, but you can for instance refere to section 3.4 in https://proceedings.neurips.cc/paper_files/paper/2022/hash/46f0114c06524debc60ef2a72769f7a9-Abstract-Conference.html. Nevertheless I ackowledge the contribution of using auxiliary functions to compute the a posteriori error. As mentionned already, I underestimated this part.
> >
> > tbc.

---

> > > ### Comment · Reviewer_8fDV · 2024-12-02
> > >
> > > continuation.
> > >
> > > > Firstly, as Table 3 demonstrates, our methods are faster than those based on residuals, negating the need for "expensive computational corrections." Secondly, NTK analysis is irrelevant to our research, which focuses on a posteriori error estimation, not training dynamics.
> > >
> > > There has been some misunderstanding on this point. I didn't mean to imply that your method is slower than conventional methods. Quite the opposite, as Table 3 shows. What I meant was that the 2nd-order methods I mentioned are much more expensive to train than the 1st-order methods, and that the loss change you propose could address this problem. Indeed, the major difficulty in training PINNs comes from the fact that the residual is ill-conditioned due to the operator (see for instance https://arxiv.org/abs/2402.01868). This is in contrast with classical losses in ML, which explains why order one optimizer work in those cases, while failing for PINN losses. Now, by taking a loss that bounds the L2 error as you do, one may expect that the loss is better conditioned, since L2 error is. That is why I suggested to you than you analyze the NTK spectrum, since this gives essential information on the problem conditioning (even if I agree that your work does not bound to training). In my opinion this is not irrelevant, since it may explain why the training gets improved.
> > >
> > > ---
> > >
> > > To sum up :
> > > - I acknowledge that I underestimated the value of the word wrt a posteriori error calculation.
> > > - I am still convinced that connections with Natural Gradient methods are missing, especially if the authors want to claim result concerning accuracy. Furthermore this could certainly be fruitful, since at the very core of natural gradient lies the fact that PINNs problem is ill conditioned, something that the loss the authors introduced seems to improve (but empirical evidences of it are missing).

---

### Meta-Review · Area_Chair_q5JE · 2024-12-22

**Metareview:**

The work is motivated from the observation that PINN losses may be unsuitable as an approximation/bound of the L2 distance between true and estimated solution. The work proposes new losses which are able to bound the L2 distance and some empirical evidence is provided supporting the effectiveness of the proposed losses. The reviewers acknowledge the progress reported, but several concerns were raised from all reviewers, including limited clarity of exposition, limited comparison with alternative approaches, e.g., which better handle the poor conditioning, limited empirical  evidence towards convincingly showing the efficacy, among others. The authors have responded to the concerns raised, resolved some, but certain concerns persist.

**Additional Comments On Reviewer Discussion:**

The reviewers acknowledged the responses from the authors, and certain differences persisted even after the discussion.

---

### Decision · Program_Chairs · 2025-01-22

Reject